

# Insights into the Australian mid-Holocene climate using downscaled climate models

Andrew L. Lowry[1], Hamish A. McGowan[1]

[1]School of the Environment, The University of Queensland, Brisbane, QLD 4110, Australia

*Correspondence to*: Andrew L. Lowry (Andrew.Lowry@uq.net.au)

**Abstract**

The mid-Holocene climate of Australia and the equatorial tropics of the Indonesian–Australian monsoon region is investigated using the Community Earth System Model (CESM) and the Weather Research and Forecasting (WRF) model. Each model is used to simulate the pre-industrial (1850) and the mid-Holocene (6000 years before 1950) climate. The results of these four

simulations are compared to existing bioclimatic modelling of temperature and precipitation. The finer resolution WRF simulations reduce the bias between the model and bioclimatic data results for three of the four variables available in the proxy dataset. The model results show that temperatures over southern Australia at the mid-Holocene and pre-industrial period were similar, and temperatures were slightly warmer during the mid-Holocene over northern Australia and into the tropics, compared to the pre-industrial. During the mid-Holocene precipitation was generally reduced over northern Australia and in the

Indonesian–Australian monsoon region, particularly during summertime. The results highlight the improved value of using finer resolution models such as WRF to simulate the palaeoclimate.

## 1 Introduction

The Holocene is the current geological epoch that started following the Younger Dryas and cessation of glacial conditions 11,700 ka BP (kilo-annum before present). The mid-Holocene corresponds to 6000 ka BP and was a time when sea levels and

greenhouse gas concentrations were roughly like present-day conditions. However, solar insolation was different due to changes in the Earth's orbital parameters (Berger and Loutre, 1991). The largest difference was the latitudinal and seasonal distribution of solar irradiance stemming from the larger obliquity and the timing of perihelion, which occurred at the austral spring equinox compared to near the austral summer equinox in present-day conditions (Otto-Bliesner et al., 2020).

These small but important differences in climatic conditions combined with reasonable repositories of proxy palaeoclimate

evidence are why the mid-Holocene has been a baseline experiment as part of the Palaeoclimate Modelling Intercomparison Project (PMIP), and has been the subject of extensive analysis using Global Circulation Models (GCM) and Earth System Models (ESM) (Joussaume et al., 1999; Braconnot et al., 2007; Brierley et al., 2020). From the large collection of simulations of the mid-Holocene climate within the PMIP framework, there has been comprehensive analysis of the global conditions, which continues to improve as models become more sophisticated (Brierley et al., 2020). Associated with this analysis, the



mid-Holocene climate of Australia and the monsoon tropics to the north have also received attention, as part of global analyses (Brierley et al., 2020), monsoon dynamics (D'Agostino et al., 2020; Jiang et al., 2015; Zhao and Harrison, 2012), and over drylands (Liu et al., 2019), using the ensemble of simulations produced through the PMIP framework. The consensus of this prior work was that the mid-Holocene over Australia was slightly cooler, by up to 0.3 °C (Brierley et al., 2020). The cooling was more intense in the austral summer, by up to 1 °C, but mildly warmer in the austral winter, although not over the whole

continent with a mild cooling in the south (Brierley et al., 2020). Monsoon precipitation in the Australian tropics was likely reduced by up to 200 mm year$^{-1}$ (D'Agostino et al., 2020) and shifted equatorward during the mid-Holocene (Jiang et al., 2015). Within the interior of Australia there was very little difference in the extent of dryland environments between the mid-Holocene and present-day, although there was a slight expansion in "arid" conditions over the central and north-west of the continent (Liu et al., 2019).

The summation of the mid-Holocene Australian palaeoclimate from the modelling community contrast with bioclimatic analyses of the mid-Holocene summarised in the OZ-INTIMATE series (Reeves et al., 2013a). In this series Reeves et al. (2013a), found that the early to mid-Holocene experienced the warmest temperatures before they declined in the late Holocene, and that precipitation likely peaked by the mid-Holocene before decreasing and becoming more variable, linked to intensification of El Ninõ, in the late Holocene. Tropical precipitation was believed to have been slightly reduced during the

mid-Holocene linked to the northward propagation of the Intertropical Convergence Zone (Reeves et al., 2013a). Herbert and Harrison (2016) developed a pollen-based palaeoclimate database for Australia, following the seminal work of Bartlein et al. (2011). This database has since been expanded to include 239 sites across Australia (pers. comm.) which provides a basis for palaeoclimate model–data intercomparison.

Using quantified climate proxy data can help to inform interpretation of the climate modelling results and provide a means of

assessing model skill in representing past climates. This kind of analysis, while extensive over Northern Hemisphere climates (e.g. Braconnot et al., 2012) where there is data (Bartlein et al., 2011), has not been tested over Australia. This presents an obvious opportunity to simulate the palaeoclimate of Australia and analyse this in relation to the newly developed pollen proxy dataset from Herbert and Harrison (2016).

In addition to the identified gap in model–proxy data intercomparison research, there is further possibility to improve the

simulated climate of Australia using downscaled models. Using regional climate models (RCM) to simulate the present-day climate is widespread, although less so to simulate the palaeoclimate (Ludwig et al., 2019). Simulations of the mid-Holocene palaeoclimate have only been performed over Europe (Russo et al., 2022; Strandberg et al., 2022, 2014; Armstrong et al., 2019; Russo and Cubasch, 2016; Brayshaw et al., 2011), North America (Diffenbaugh and Sloan, 2004; Diffenbaugh et al., 2003), Asia (Paeth et al., 2019; Kim and Kim, 2014; Yu et al., 2014; Polanski et al., 2012; Liu et al., 2010; Zheng et al., 2007,

2004; Kim et al., 2005), Africa (Patricola and Cook, 2007), and New Zealand (Ackerley et al., 2013). There is a consensus that some improvement in simulated climate can be achieved by using finer resolution models, particularly hydrological processes (Armstrong et al., 2019). These improvements stem from the better depiction of feedback and physical processes at the regional scale (Armstrong et al., 2019; Ludwig et al., 2017). Here we analyse the mid-Holocene palaeoclimate of Australia





using both an ESM and RCM to evaluate the skill of both models with respect to bioclimatic modelled proxy data. Numerical

modelling also allows the palaeoclimate of large regions of Australia to be simulated where palaeo bioclimatic data are scarce or unavailable (Reeves et al., 2013a). These simulations provide the first downscaled palaeoclimate analysis of the mid-Holocene in Australia and insight into the climate that would have been experienced by early human populations.

## 2 Data and Methods

To investigate the palaeoclimate of the mid-Holocene (MH) two simulations were performed: pre-industrial (1850 Common

Era) control (PI), and 6 ka BP. These two simulations were performed with the Community Earth System Model (CESM) (Hurrell et al., 2013), which were then downscaled over an Australian domain using the Weather Research and Forecasting model (WRF) (Skamarock et al., 2021). The greenhouse gas concentrations and orbital parameters used in each simulation were modified and are shown in Table 1. The radiative forcing anomaly resulting from the slight differences between the MH and PI greenhouse gas concentrations is 0.54 W m$^{-2}$ (Etminan et al., 2016). The orbital parameters affect the seasonal and

latitudinal distribution, and magnitude of insolation at the top of the atmosphere. At MH perihelion occurred near the austral spring equinox, compared to slightly after the austral summer solstice at PI. This change in timing of perihelion combined with the small differences in obliquity and eccentricity, resulted in strong positive insolation anomalies during the austral winter through to the austral spring over Australia, followed by negative anomalies during the austral summer and autumn (Fig. 1).

| Simulation | $CO_2$ (ppm) | $CH_4$ (ppb) | $N_2O$ (ppb) | Obliquity (°) | Eccentricity | Angular Precession (°) |
|---|---|---|---|---|---|---|
| PI | 284.7 | 791.6 | 275.68 | 23.459 | 0.016764 | 280.33 |
| MH | 264.4 | 597.0 | 262.0 | 24.105 | 0.018682 | 180.87 |

**Table 1: The boundary conditions and orbital parameters for the pre-industrial and mid-Holocene simulations.**



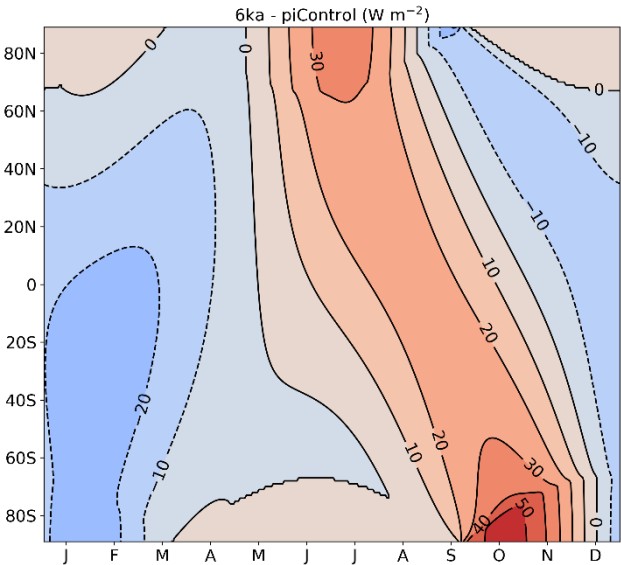

Figure 1: Seasonal cycle of insolation anomalies for mid-Holocene minus pre-industrial (W m⁻²).

## 2.1 Earth System Model

The Community Earth System Model (CESM) version 1.2 (Hurrell et al., 2013) was used to provide the coarse resolution lateral boundary conditions for the regional climate modelling. CESM is a fully coupled earth system model that used the Community Atmosphere Model v5.3 with 30 atmospheric layers, and Community Land Model v4.0, at a horizontal resolution of 1.9° latitude by 2.5° longitude. The ocean and ice components were the Parallel Ocean Program and Community Ice Code at nominal 1° latitude and longitude. The PI and MH simulations were performed for 31 years, starting from climatically stable restart positions provided by the National Center for Atmospheric Research (NCAR), using only the last 30 years for analysis. To ensure the model was in equilibrium we checked the residual top of the model radiation balance, which was 0.11 W m⁻² for PI and 0.12 W m⁻² for MH. CESM and its predecessor the Community Climate System Model (CCSM) have been shown to provide suitable simulations of the palaeoclimate (Brady et al., 2013; Otto-Bliesner et al., 2020) and is representative of other models that have participated in the PMIP (Brierley et al., 2020; Harrison et al., 2015). CESM provided 6-hourly atmospheric boundary condition data, daily sea surface temperature and sea ice cover, and monthly average surface data required by the regional climate model.



## 2.2 Regional Climate Model

The downscaled regional climate model used was the WRF model version 4.4 (Skamarock et al., 2021). The WRF model is a dynamic climate model that solves the non-hydrostatic Euler equations with terrain following vertical coordinates (Skamarock et al., 2021). WRF has been developed by NCAR and used extensively for research and commercial applications in numerical weather prediction (Skamarock et al., 2021). In palaeoclimate modelling WRF has been used successfully in Europe (Russo et al., 2024; Ludwig and Hochman, 2022; Velasquez et al., 2022, 2021; Stadelmaier et al., 2021; Ludwig et al., 2021; Pinto and Ludwig, 2020; Schaffernicht et al., 2020; Ludwig et al., 2018, 2017), and Asia (Yu et al., 2018, 2014; Yoo et al., 2016) to downscale ESM/GCMs during the mid-Holocene and Last Glacial Maximum.

The WRF domain was set up at 50 km resolution, with 45 vertical eta levels up to 30 hPa, centred over Australia and included the equatorial tropics to capture monsoon processes (Fig. 2). The WRF model was run, using the CESM output as boundary and initial conditions, for 31 years, with the first year discarded as spin-up. The model domain included a relaxation zone of 10 grid cells, which were also discarded from the analysis. The physics schemes used are shown in Table 2, which were modified to account for changes in greenhouse gas concentrations and solar irradiance for each simulation (Table 1). The cumulus scheme was modified to suit tropical conditions by setting the factor Fs=0.6 from the humidity reference profile as prescribed by Janjic (1994), evaluated by Fonseca et al. (2015) for the Maritime Continent, and by Evans et al. (2012) for Australia. Sub-grid scale feedback between the cumulus parameterisation and radiation schemes was enabled (Koh and Fonseca, 2016). To improve model estimates of the diurnal cycle of skin temperature over water the parameterisation of Zeng and Beljaars (2005) was used. The monthly values of albedo, leaf area index, fraction of photosynthetically available radiation, and deep soil temperature were supplied from the Community Land Model taking the average of the last 30 years of the CESM simulation. The vegetation and land use in WRF were prescribed for PI and MH from the simulations of Allen et al. (2020), which were mapped to the 28 category United States Geological Survey (USGS) classification available in WRF (Table S1) at 0.5° resolution. The vegetation simulations from Allen et al. (2020) include the land–water distribution from the ICE-5G model of Peltier et al. (2004) at 0.1° resolution. The only inland lake resolved at 0.5° resolution was Kati Thanda–Lake Eyre, which was reclassified from ocean to lake and the surface water temperature was estimated from daily average surface air temperature.

## 2.3 Proxy Data

The model simulations were evaluated against a new pollen-based reconstruction of seasonal temperature, mean annual precipitation, and moisture availability (Herbert and Harrison, 2016). This dataset was developed to fill the gap in the widely used pollen-based reconstruction from Bartlein et al. (2011), which has no data points in Australia. The original pollen reconstruction from Herbert and Harrison (2016) has been updated to include many more data locations (pers. comm.) totalling 239. Unfortunately, due to high correlations between mean annual temperature and the two seasonal temperature variables in the modern climate, it was not possible to include all three temperature variables, and Herbert and Harrison (2016) have only



provided reconstructions for the following variables: mean temperature of the warm month (MTWA), mean temperature of the cold month (MTCO), mean annual precipitation (MAP), and the plant available moisture index (α). In addition to this continental scale proxy data set there are four individual sites that have been quantifiably analysed: Eagle Tarn (42.6799°S,
146.5914°E), and Platypus Tarn (42.6734°S, 146.5868°E) in Tasmania (Rees and Cwynar, 2010), Lake McKenzie (25.4475°S, 153.0533°E) on Fraser Island, Queensland (Woltering et al., 2014), and Swallow Lagoon (27.4986°S, 153.4547°E) on Stradbroke Island, Queensland (Barr et al., 2019). The two sites on Tasmania show temperatures for the warm month in the MH were slightly cooler than present-day temperatures (Fig. 5, Rees and Cwynar, 2010). In contrast, the MAT on Fraser Island in Queensland showed temperatures were 0.9 °C warmer at 5.8 ± 0.3 ka BP relative to 1950 (Woltering et al., 2014). On
Stradbroke Island in Queensland MAP was higher by around 500 mm year$^{-1}$ during the MH and much less variable than late Holocene and present-day conditions (Barr et al., 2019).

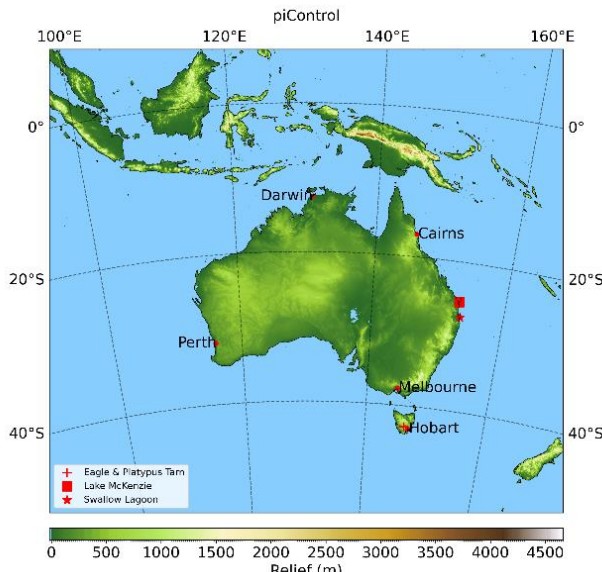

**Figure 2: The topographical height from ETOPO1 for the WRF model domain. Cities and specific proxy sites mentioned in the text are shown.**


| Physics | Scheme Name | Reference |
|---|---|---|
| Boundary Layer | Mellor-Yamada-Janjic | (Janjic, 1994) |
| Cumulus | Betts-Miller-Janjic | (Betts, 1986; Betts and Miller, 1986; Janjic, 1994) |
| Land-Surface Layer | Unified Noah Land Surface Model | (Tewari et al., 2004) |
| Microphysics | Double-Moment 5 Class | (Lim and Hong, 2010) |
| Radiation | Rapid Radiative Transfer Model for GCMs | (Iacono et al., 2008) |
| Surface Layer | Eta | (Janjic, 1994) |

**Table 2: The physics schemes used in the WRF simulations.**



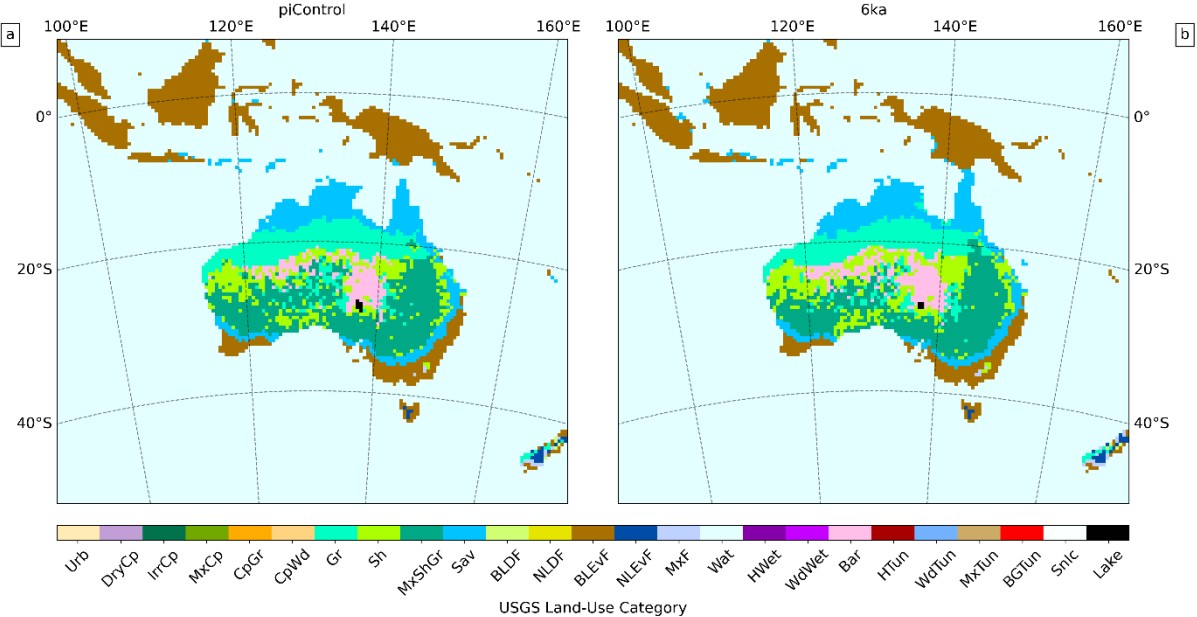

**Figure 3: Vegetation distribution used in the WRF simulations for pre-industrial (a) and mid-Holocene (b). See Table S1 for a description of the codes.**

## 3 Results and Discussion

The results of the WRF simulations are first validated against present-day conditions, followed by a discussion of the mean state of the PI and MH simulations, and the differences between them. Where possible the climate anomalies are compared to available proxy data and are re-gridded to a common rectilinear grid for analysis. Climate models use a modern calendar for monthly analysis, but the length of each month changes as the transit speed of the Earth's orbit varies with the timing of perihelion. Sub-annual model results were adjusted for the variations in these month lengths using the PaleoCalAdjust tool developed by Bartlein and Shafer (2019).

### 3.1 Pre-Industrial Climate Conditions

The results of the PI simulation were validated against present-day reanalysis and satellite products. These comparisons were complicated by the differences between 1850 and late 21[st] century conditions, but nevertheless provide a basic benchmark for the WRF simulations. To validate MAT from the PI simulation we obtained the ANUCLIM dataset which provides monthly temperature data on a 0.05° grid over continental Australia (Xu and Hutchinson, 2011). After re-gridding the WRF and ANUCLIM data to a rectilinear 0.45° grid the RMSE between the WRF PI simulation and ANUCLIM data was 0.29 °C (Fig.



S1). The temperature distribution was very similar between the PI simulation and ANUCLIM, with annual temperatures in the tropical north of Australia 27.6 °C at Darwin and 22.2 °C at Cairns, transitioning to cooler conditions in the south, 18.1 °C at
Perth, 13.8 °C at Melbourne, and 11.0 °C at Hobart (Fig. 4a). Validation of precipitation was done with reference to the Tropical Rainfall Measuring Mission (TRMM) satellite dataset (Huffman et al., 2010). The TRMM and WRF MAP data were re-gridded to the same rectilinear 0.45° grid and the RMSE between the two was 0.5 m year$^{-1}$ (Fig. S2). The differences between WRF and TRMM were heterogeneous (Fig. S2), with WRF simulating heavier precipitation in tropical locations over water, but generally lighter precipitation over tropical land. Over mid-latitude continental Australia WRF simulated lighter
precipitation over the coastal locations south of Perth, but over most of the remaining continental land WRF simulated heavier precipitation. The annual precipitation was 1233 mm year$^{-1}$ at Darwin, 772 mm year$^{-1}$ at Cairns, 480 mm year$^{-1}$ at Perth, 877 mm year$^{-1}$ at Melbourne, and 659 mm year$^{-1}$ at Hobart. Overall, the spatial distribution of precipitation was well captured with the meridional gradient from the tropics to the Southern Ocean in the TRMM dataset replicated in the WRF simulation. These small differences between WRF and the reanalysis and satellite datasets indicate that WRF is suitable for Australian
palaeoclimate simulations.

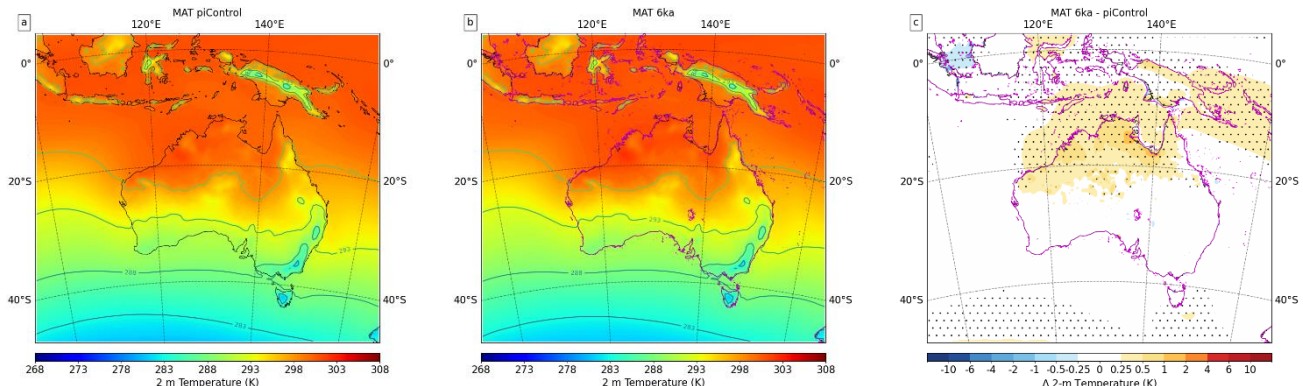

**Figure 4: 2 m Temperature for the WRF pre-industrial simulation (a), mid-Holocene simulation (b), and the difference mid-Holocene minus pre-industrial (significant changes, two-sided Student's t-test, 95% confidence interval, are indicated by dots) (c). The purple line in (b) and (c) is the coastline in the mid-Holocene.**

**3.2 Mid-Holocene Climate Conditions**

The simulated MAT in the MH shows that conditions were generally very similar to PI over the Australian domain (Fig. 4). The positioning of the MAT isotherms in the MH and PI simulations are nearly identical between the two simulations (Fig. 4a,b). Compared to the PI the MH MAT was within ± 0.25 °C over most of continental Australia and in the north-west of the model domain over Borneo, Sumatra, and Java. Over the north of Australia and into the tropical seas to the north and north-
east of the Australian continent temperatures were between 0.25–1 °C warmer in the MH. The standard deviation of MAT is predominantly below 1 °C over most of the domain, with a small area in The Kimberley and Pilbara getting up to 1.3 °C (Fig.



S3a). During the MH the MAT at Darwin was 28.2 °C, 22.4 °C at Cairns, 18.0 °C at Perth, 13.8 °C at Melbourne, and 11.0 °C at Hobart.






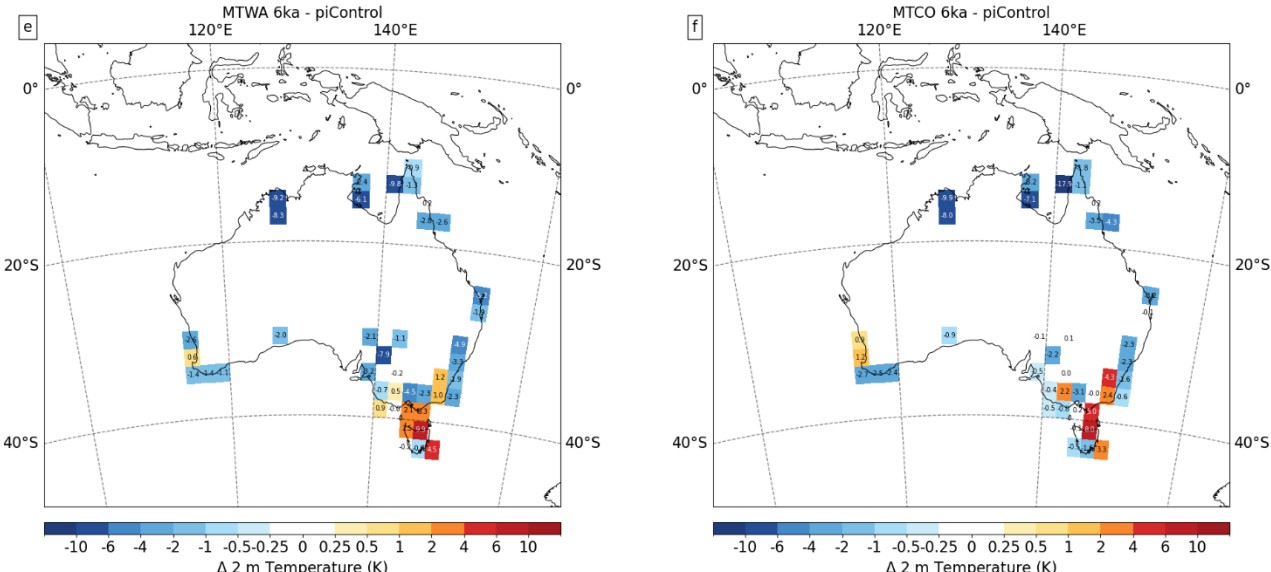

**Figure 5: Seasonal 2 m temperature differences between mid-Holocene and pre-industrial for MTWA (left) and MTCO (right). The top row is the WRF simulations (a) and (b). The middle row is the CESM simulations (c) and (d). The bottom row is the pollen proxy dataset (Herbert and Harrison, 2016) at 2° resolution (e) and (f). In (a), (b), (c), and (d) the purple line is the coastline in the mid-Holocene, and significant changes, two-sided Student's t-test, 95% confidence interval, are indicated by dots.**


The January temperatures over the Australian continent were predominantly cooler during the MH, on average 0.35 °C cooler over Australia (Fig. 5a) although not statistically significant at the 95% level, and during July in the north-west of Australia temperatures were slightly warmer, with a continental average of 0.19 °C warmer (Fig. 5b). These continental conditions were in response to the positive insolation anomalies in the austral winter and early spring, and negative anomalies in the austral

summer and autumn (Fig. 1). There was, however, a large range of year-to-year fluctuations of seasonal temperatures over continental Australia in both the MH and PI, such that on any given year conditions could have been warmer or cooler in the MH relative to PI (Fig. S3c,d). The oceanic response to insolation, however, was the opposite with warmer tropical and Southern Ocean conditions in January. The warmer conditions over tropical Australia and into the Maritime Continent in January match the response from the coarser resolution ESM where temperatures over the tropical latitudes were 0.25–1 °C

warmer in the MH (Fig. 5c). The oceanic response reflects the higher inertia of water relative to land, that mutes the temperature response due to the changing insolation forcing during the austral summer.

The proxy evidence for MH temperatures showed a cooling trend for both January and July (Fig. 5e,f), in a few locations these were significantly cooler particularly in tropical latitudes. Over south-eastern Australia the proxy evidence indicates temperatures were both cooler and warmer depending on the grid cell. In January temperature differences were larger than in

July, but there was an unclear picture that the MH was generally cooler or warmer, as the results appear to be specific to the location of the proxy record. The mean absolute error (MAE) between the proxy evidence and the two simulations was very similar at 2° resolution for both MTWA and MTCO, 2.8 °C and 2.7 °C respectively. If, however, the proxy and WRF results



are re-gridded at 0.45° resolution, then the MTWA reduces to 2.4 °C, which is an improvement relative to the coarser scale CESM result.

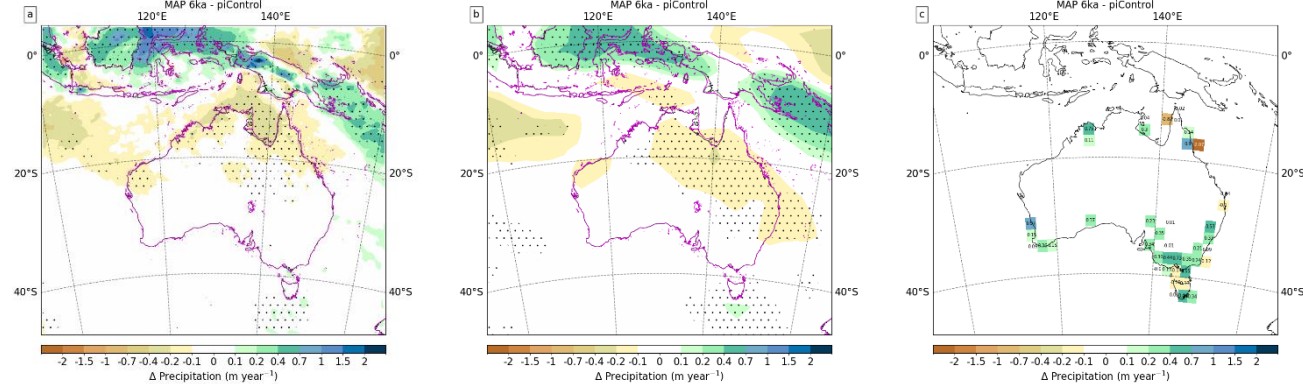

**Figure 6: Annual precipitation differences between mid-Holocene and pre-industrial, WRF (a), CESM (b), and pollen-based proxy data from Herbert and Harrison(2016) at 2° resolution (c). In (a) and (b) significant changes, two-sided Student's t-test, 95% confidence interval, are indicated by dots and the purple line is the coastline in the mid-Holocene.**

Annual precipitation during the MH was within ± 0.1 m year$^{-1}$ of PI over most of continental Australia and particularly south of 20°S (Fig. 6a). Over the tropical latitudes MAP was more heterogeneous with drier conditions over northern Australia and the north-east Indian Ocean, but wetter conditions over the equatorial islands of the Maritime Continent and the Solomon Sea. The same pattern in the tropics exists in the coarser resolution CESM results (Fig. 6b), although the magnitude of the responses was more muted, particularly over the equatorial latitudes. The WRF model at the finer resolution captured the orographic effects at these locations, particularly over New Guinea. The wetter anomalies over the equatorial Maritime Continent, Solomon Sea, and the highlands of New Guinea were statistically significant at the 95% level, stemming from the smaller relative year-to-year fluctuations at these locations (Fig. S3b). The same is true of the drier anomalies in the Arafura Sea and the Gulf of Carpentaria, where the year-to-year fluctuations were much smaller. In the north-east Indian Ocean and equatorial western Pacific Ocean, the anomalies were not statistically significant at the 95% level due to the large fluctuations in year-to-year precipitation differences (Fig. S3b). The MAP during the MH was 1023 mm year$^{-1}$ at Darwin, 546 mm year$^{-1}$ at Cairns, 442 mm year$^{-1}$ at Perth, 873 mm year$^{-1}$ at Melbourne, and 698 mm year$^{-1}$ at Hobart.

The proxy evidence for MH precipitation showed a generally wetter continent except a few individual locations. In south-east Australia, where most of the proxy data exists, precipitation was predominantly heavier in the MH up to 700 mm year$^{-1}$, which would constitute a near doubling of precipitation from modern conditions. Over Cape York Peninsula the proxy data showed both a wetter and drier climate during the MH depending on the sample location. The MAE between the proxy evidence and CESM was 352 mm year$^{-1}$, compared to 363 mm year$^{-1}$ for WRF at 2° resolution. Re-gridding the proxy and WRF results to 0.45° reduced the MAE to 294 mm year$^{-1}$, which is a 20% improvement.



**Figure 7: Seasonal precipitation differences between mid-Holocene and pre-industrial Dec, Jan, and Feb from WRF (a), Jun, Jul, and Aug from WRF (b), Dec, Jan, and Feb from CESM (c), and Jun, Jul, and Aug from CESM (d). Significant changes, two-sided Student's t-test, 95% confidence interval, are indicated by dots and the purple line is the coastline in the mid-Holocene. Note the different units compared to Fig. 6.**

During the austral summer precipitation over most of continental Australia was less during the MH, extending into the Timor and Arafura Seas, and over most of New Guinea and into the equatorial western Pacific Ocean (Fig. 7a,c). The dry anomalies over New South Wales, southern Northern Territory, and Queensland were statistically significant at the 95% level, but the year-to-year fluctuation was much larger in the north-west of the domain (Fig. S4a). Across the Indonesian Archipelago





precipitation conditions were much wetter during the MH, but only statistically significant at the 95% level in a few locations. This summertime precipitation over the Indonesian–Australian monsoon region indicates a sharp north-west south-east transition between the MH and PI. There was a strong intensification in the north-western part of the monsoon tropics, which
transitioned to drier conditions in the south-eastern part of this domain. The WRF model at the finer resolution was able to simulate much larger precipitation rates, which are illustrated in the differences in the magnitude of the anomaly in the summertime precipitation (Fig. 7a,c).

The seasonal wind anomalies along with sea level pressure (SLP) are shown in Fig. 8. Summertime SLP was reduced in the MH over most of the domain, except in the centre of continental Australia and in the south of the domain (Fig. 8a). There was
reduced westerly flow during the MH over the Indonesian Archipelago indicating weaker monsoon conditions relative to PI (Fig. 8a), which was coincident with reduced summertime precipitation (Fig. 7a,c). This was also evident in the 850 hPa zonal wind anomalies between 15°S and the equator (Fig. S6c). The 850 hPa divergence was also increased in the MH over the north-east Indian Ocean and into the Timor Sea (Fig. 9a). The spatial distribution of 850 hPa divergence differences between the MH and PI was reasonably coincident with the reduced precipitation anomalies in summertime, except a small region of
convergence around Carpentaria. There was also coincident 850 hPa convergence anomalies in the Indonesian Archipelago with increased summertime precipitation (Fig. 7a,c). The size of the monsoon convergence zone was slightly enlarged in the MH (Fig. S5a) compared to PI (Fig. S5b), although the southern extent was similar at just south of 20°S. This increased convergence zone differs over northern Arnhem Land which contributed to the reduced precipitation there from weaker westerly flow. In the extra-tropics there was also reduced westerly flow over the Southern Ocean in the MH summertime
relative to PI (Fig. 8a), coincident with increased summertime temperatures (Fig. 5a,c), but not coincident with a decline in precipitation (Fig. 7a,c).

During the austral winter precipitation differences were almost entirely within ± 0.2 mm day⁻¹ over continental Australia (Fig. 7b,d), although there was large year-to-year fluctuation of up to 2 mm day⁻¹ (Fig. S4b). Over the north-eastern Indian Ocean and across the Indonesian Archipelago into the Java and Banda Seas conditions were drier by up to 2 mm day⁻¹. Across the
equator in Borneo, the Celebes Sea, and over New Guinea precipitation was much heavier during the MH. Similar to the continental results there were large year-to-year fluctuations. The only locations where the anomalies were statistically significant (at the 95% level) were Borneo, around Tasmania, the ocean to the south-west of Australia, and a few locations on continental Australia (Fig. 7b).

The relationship between wintertime precipitation and SLP or wind anomalies was less clear than in summertime. Wintertime
SLP was predominantly higher in the MH (Fig. 8b), but there was a region of increased SLP over the southern Tasman Sea and increased westerly flow which was coincident with increased precipitation (Fig. 7b,d). In the Southern Ocean to the south of Australia there was increased SLP and weaker westerly flow which was coincident with reduced precipitation (Fig. 7b,d). The convergence zone between the south-east trades and extra-tropical westerlies was located at a similar latitude in both the MH and PI simulations at around 30°S (Fig. S5c,d) and there was negligible difference in the zonally averaged 850 hPa extra-
tropical westerlies (Fig. S6d). There was no coherent relationship between the 850 hPa divergence anomalies (Fig. 9b) and



wintertime precipitation, with the exception of the extreme north of the domain where increased convergence was coincident with increased MH precipitation (Fig. 7b,d).

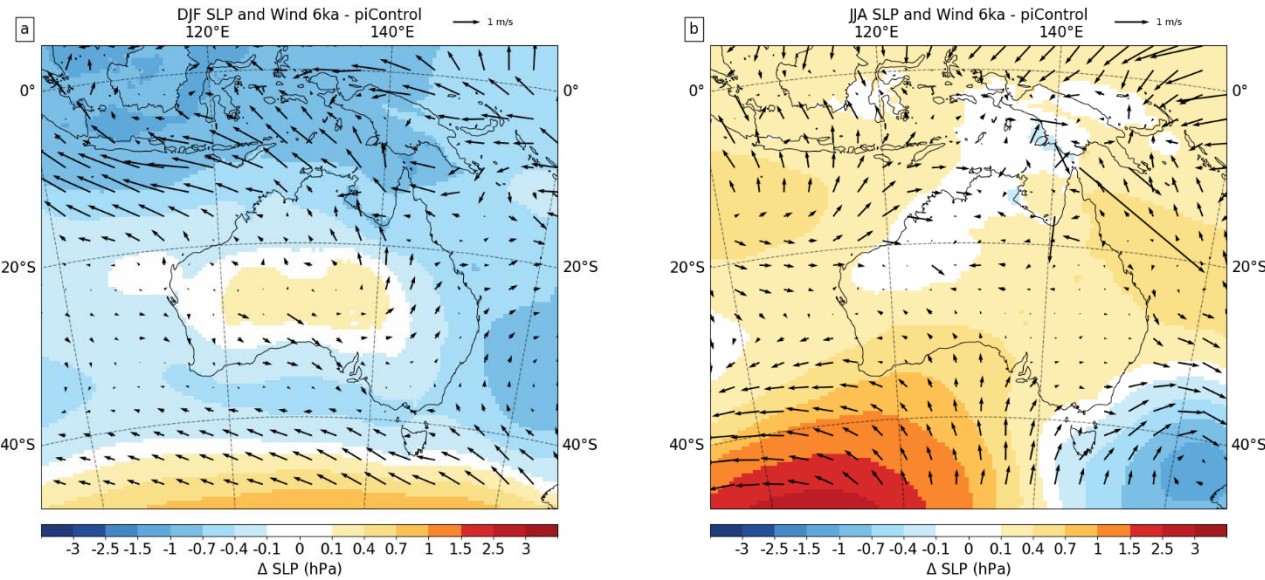

**Figure 8: Sea Level Pressure (SLP) as the coloured contours and wind speed as the vectors of the differences between mid-Holocene and pre-industrial from the WRF simulations for Dec, Jan, and Feb (a), and Jun, Jul, and Aug (b).**

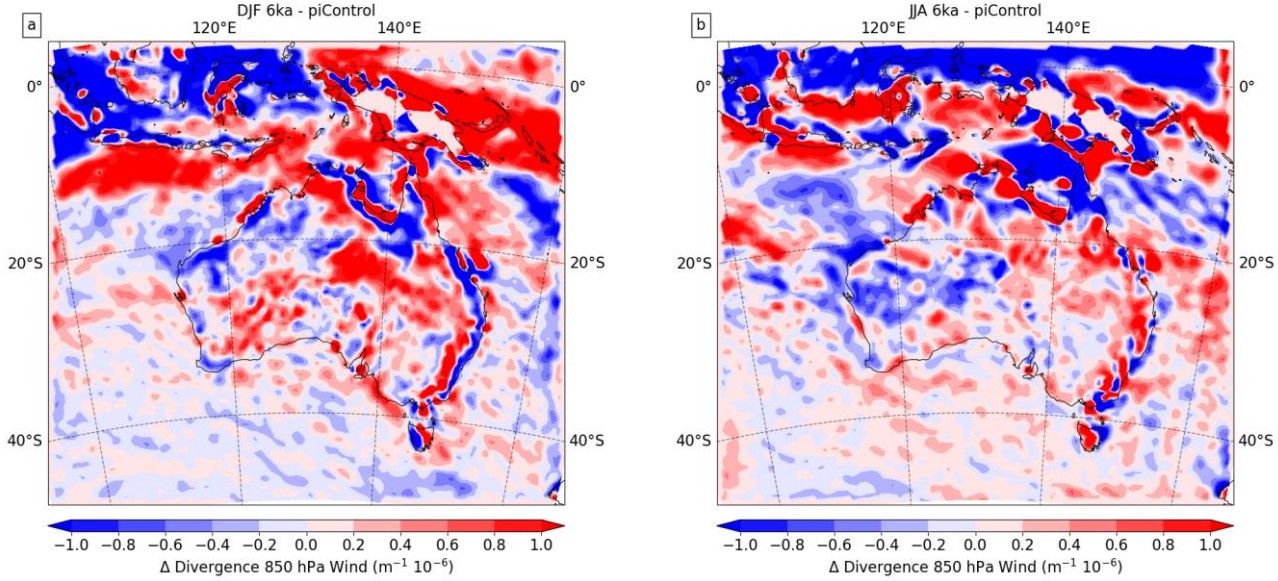

**Figure 9: 850 hPa divergence difference between mid-Holocene and pre-industrial from the WRF simulation for Dec, Jan, and Feb (a), and Jun, Jul, and Aug (b).**



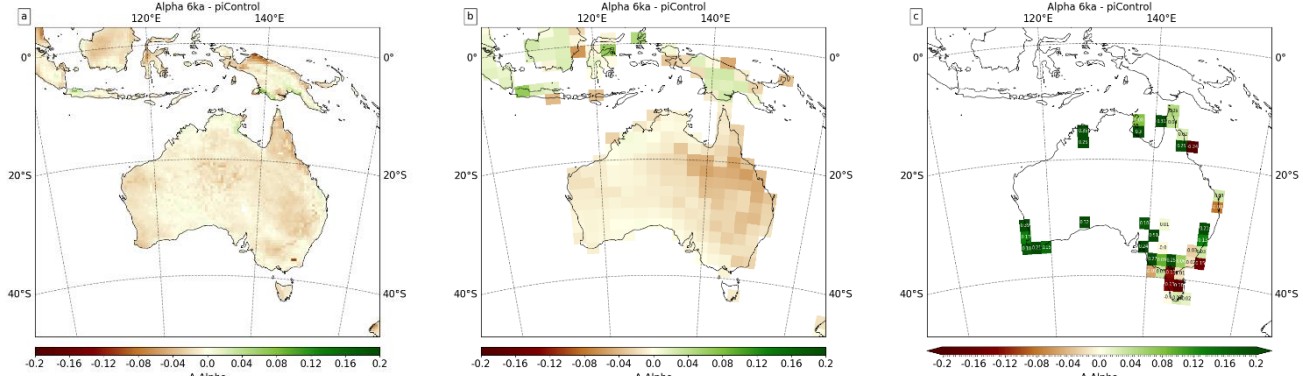

**Figure 10: Mean annual plant available water index (alpha) difference between mid-Holocene and pre-industrial for WRF (a), CESM (b), and pollen proxy data (c). The model results in (a) and (b) are masked for cells where ≥ 25% of the cell is land.**

Alpha is the ratio of actual evapotranspiration to equilibrium evapotranspiration on an annual basis and can indicate the amount of growth limiting drought stress on plants (Prentice et al., 1993). In both the WRF and CESM simulations alpha was slightly lower during the MH over continental Australia. In the WRF simulation there was a small region of positive anomaly in western Carpentaria which was the location of warmest MAT differences (Fig. 4c). In the tropics to the north of Australia there were two locations of positive alpha anomalies in the WRF simulation, on Java and southern New Guinea, which coincided with locations of increased precipitation (Fig. 6a). In contrast to this however, in the north of New Guinea there was a location of large decreased alpha where precipitation was also increased during the MH. The CESM simulation indicates positive alpha anomalies in the MH in the tropics to the north of Australia, in contrast to the WRF simulation (Fig. 10b), which were coincident with the positive precipitation anomalies (Fig. 6b).

The lower alpha anomalies over continental Australia contrast with most of the proxy evidence (Fig. 10c), which shows mostly positive anomalies although there were some grid cells in south-eastern Australia that show negative anomalies. The MAE between the proxy data and CESM simulation was 0.11, which was higher than the WRF simulation at 0.08 at 2° resolution and improves when the proxy data was compared to the WRF simulation at 0.45° resolution with a MAE of 0.06.

**4 Discussion**

The previous section outlined the changes in climate over the Australian region at the MH compared with PI conditions from the coarser resolution simulation of CESM and the finer resolution of WRF, and how these two simulations compared with results from bioclimate modelling using proxy evidence. This section will summarise these findings and compare them to other work.



## 4.1 Temperature

The MAT in both CESM and WRF were largely in agreement, showing statistically significant warming of 0.25–1 °C over central Australia into the northern tropics and extending into the seas between Indonesia, New Guinea, and Australia (Fig. 4c). These warmer temperatures contrast with the multi-member ensemble of PMIP3 and PMIP4 anomalies for the MH that found a 0–0.3 °C cooling over continental Australia (Brierley et al., 2020) and the 0.25–0.5 °C cooling from CESM2 (Otto-Bliesner et al., 2020). The spread of results from the multi-member ensemble (Brierley et al., 2020) suggest that some models simulated a warming over Australia, but this would still predominantly fall below the results presented here. Despite this the qualitative summary from the OZ-INTIMATE project found that there was a thermal expansion of the Indo–Pacific Warm Pool (IPWP) during the MH (Reeves et al., 2013b), which would agree with the temperatures found in Fig. 4c. The one annual temperature record from the proxy literature found the temperature on Fraser Island was 0.9 °C warmer in the MH (Woltering et al., 2014), which was above that found in either CESM (not shown) or WRF (Fig. 4c).

The austral winter temperatures in Australia were warmer or the same in the MH with a few coastal and oceanic locations showing slightly cooler temperatures, in both the CESM and WRF simulations (Fig. 5). The CESM simulation found that the warmer winter temperatures traversed the width of the continent, whereas in the WRF simulation the warmer temperatures were only in the central and north-west of Australia. WRF also simulated cooling over Tasmania and the Gulf of Carpentaria not found in the CESM simulation. The multi-member ensemble of PMIP3 and PMIP4 simulations found a warming over continental Australia (Brierley et al., 2020) in agreement with the CESM simulation and mostly in agreement with the WRF simulation. The CESM2 simulation found winter temperatures were ± 0.25 °C over continental Australia (Otto-Bliesner et al., 2020). Over oceanic grid cells the PMIP3 and PMIP4 simulations were cooler during the MH which contrasts with the CESM and WRF simulations. The spread of results in the multi-member ensemble (Brierley et al., 2020) suggests that some models simulated conditions found in the CESM and WRF simulations.

The austral summer temperatures in Australia were cooler over the sub-tropical and mid-latitude locations in the WRF simulation, and in a slightly less uniform manner in the CESM simulation (Fig. 5). In the tropics and to the south of continental Australia temperatures were warmer in the MH. Over continental Australia a similar result was found in the multi-member ensemble from PMIP3 and PMIP4 results with a cooling of 0.3–1.2 °C (Brierley et al., 2020). Over oceanic grid cells the multi-member ensemble found summer temperatures were cooler in the MH by 0–0.3 °C relative to PI (Brierley et al., 2020), which contrasts with the tropical and Southern Ocean results found in both the CESM and WRF simulations. The WRF and CESM results were more closely aligned with the CESM2 simulation from Otto-Bliesner et al. (2020), who found cooler land temperatures, but tropical and mid-latitude ocean temperatures were ± 0.25 °C. There were two proxy records of warm month MH anomalies from Tasmania, both showing conditions were slightly cooler during the MH (Fig. 5, Rees and Cwynar, 2010), which contrasts with the warmer temperatures found in both the CESM and WRF simulations.

The MH temperatures from the pollen proxy dataset, showed a cooling in the MH over northern Australia in the austral winter and summer (Fig. 5e,f). Over the southern latitudes of Australia, the proxy records show some locations with cooling in the



MH and in others warming depending on the exact grid cell; these results exist in both the warm and cold months. The temperature anomalies shown in Fig. 5e,f were re-gridded to 2°, but the extreme values exist at lower resolutions as well. There

are multiple potential causes of these extreme values discussed in Herbert and Harrison (2016). The first is that to improve the number of sites included in the dataset, surface conditions from a gridded dataset (i.e. ANUCLIM) were used for all sites, not core tops, as not all sites have a core top value. The justification for this was that this increased the number of sites that can be included in the dataset, which outweighed the use of only sites with core top values (Herbert and Harrison, 2016). The downside was that site specific information was lost from the core top, which would reduce some of the extreme cases. A second

contributing factor was when there was pollen identified that have few modern analogues, this can lead to large biases in the results. Despite these concerns the dataset is a valuable reference from which modelling results can be compared. The results presented here show that there was an improvement in using the finer resolution results for modelling of warm month anomalies compared to the coarser results from CESM. The same was not evident in the cold month results, but there was nevertheless no detriment in model–proxy comparison and the increased resolution facilitates identification of some fine scale effects in

coastal and locations of steep orography that are not captured in the coarser model results.

## 4.2 Precipitation and Moisture

The MAP during the MH was slightly reduced over tropical latitudes of continental Australia in the WRF simulation, and in the north and east of Australia in CESM (Fig. 6a,b). Over the remainder of Australia precipitation was the same during the MH and PI. In the equatorial tropics annual precipitation was heavier over Borneo, New Guinea, and the islands of the Banda

Arc. To the immediate north of Australia precipitation was reduced in the MH, which extended onto the north of the continent in the WRF simulation. There was one site on Stradbroke Island in Queensland that found MAP was higher by around 500 mm year[-1] during the MH (Barr et al., 2019), a result not replicated in the WRF or CESM simulations. The bioclimatic evidence from the OZ-INTIMATE series found that river discharge was enhanced in temperate climates (Petherick et al., 2013), but that conditions were more arid in the interior of the continent (Fitzsimmons et al., 2013). In the Australian tropics there was a

reduction in fluvial activity and re-activation of dunes in the Gregory Lakes in north-western Australia (Reeves et al., 2013b), mangrove contraction from 7.4 ka BP (Proske et al., 2014), and reductions in rainforest taxa (Field et al., 2017) in the Kimberley. There was a slight equatorward shift of the Intertropical Convergence Zone and contraction of the extent of the IPWP, resulting in a small reduction in monsoon activity over northern Australia (Reeves et al., 2013b). These reductions in northern Australian precipitation agree with the findings in the WRF and CESM simulations.

To provide a more quantifiable analysis between the modelling and the bioclimatic results, the simulations were compared to the pollen proxy dataset. The pollen proxy dataset also found precipitation was generally increased in the MH (Fig. 6c) except a couple of locations around Cape York Peninsula and the east and south-east of Australia. The bias between the proxy dataset and the simulations was similar at 2° but improved by 20% for the WRF simulation when the resolution was reduced to 0.45°. The moisture availability variable (α) also had a reduction in bias in the WRF simulation compared to the CESM. These results





highlight the benefit of finer resolution models, over and above the better representation of changes in response to feedback and physical processes.

On a seasonal basis austral winter precipitation was similar between MH and PI over continental Australia. The largest differences were evident in the intensification of equatorial precipitation contrasting with a reduction over the Java Sea, Banda Sea, and the north-east Indian Ocean. The same differences in tropical wintertime precipitation were found in the CESM2

simulation from Otto-Bliesner et al. (2020) and the multi-model ensemble of PMIP3 and PMIP4 simulations (Brierley et al., 2020). The overall wintertime precipitation distribution for the whole domain matches that from CESM2 and the multi-model ensemble.

The summertime precipitation is dominated by the Indonesian–Australian monsoon, although the level of dryness in the south of Australia can influence drought and forest fires. The summertime precipitation in the higher latitudes of Australia was very

similar between MH and PI, although the WRF simulation found slightly wetter conditions over Tasmania and eastern Victoria. In the tropical north of Australia summertime precipitation was reduced in the MH in both the CESM and WRF simulations as far north as the Indonesian Archipelago. A similar trend was found in the multi-member ensemble, although the reduction in precipitation was more severe, up to 0.8 mm day$^{-1}$ (Brierley et al., 2020). The CESM2 simulation showed only continental Australian grid cells experienced reductions at the MH, which transitioned to increased precipitation in the Timor and Arafura

Seas (Otto-Bliesner et al., 2020). The reductions in summertime precipitation were considered to be a result of a response to the timing of perihelion, which shifted to the austral spring equinox during the MH and resulted in a northward propagation of the summertime precipitation domain (Brierley et al., 2020; Jiang et al., 2015). In a different multi-model analysis focussing on the mid-Holocene Southern Hemisphere monsoon D'Agostino et al. (2020) found that the northern Australian monsoon extent over land grid cells contracted 13.8% and the precipitation was 10.5% lower in the MH. D'Agostino et al. (2020) used

moisture budget decomposition to show that the reductions in MH monsoon activity were primarily dynamical weakening, i.e. stronger anticyclonic conditions and westward shift of the main Walker circulation updraught. The WRF simulation showed a similar weakening of westerly winds over the north-eastern Indian Ocean and slightly increased SLP over the middle of the continent (Fig. 8a), which reduced onshore flow on the western coastline of tropical Australia and the resultant decrease in summertime precipitation.

**5 Conclusions**

This research presents the first evaluation of downscaled climate modelling of the MH climate over Australia and the surrounding region. This evaluation consisted of simulating the MH and PI climate, with an ESM and RCM, and analysing both simulations against the available proxy records. These results offer the highest resolution analysis of the MH climate over Australia. The model results show that there was little difference in temperature over southern Australia between the MH and

PI, and over northern Australia and into the tropics temperatures were slightly warmer during the MH relative to PI. Precipitation during the MH was generally reduced over northern Australia and in the Indonesian–Australian monsoon region,





particularly during summertime months. The results of the coarser ESM and finer RCM were generally in agreement, however significantly greater granulation was evident in the finer resolution WRF simulations in response to depiction of feedback and physical processes. The comparison of the two simulations to the pollen proxy dataset shows that there was some improvement

to be gained from the finer resolution model. These improvements were reductions in MAE between the model and proxy data of MTWA, MAP, and the moisture availability index.

Further analysis is required to investigate the drivers of changes in the monsoon by decomposing the moisture budget into its thermodynamic and dynamic components. Such an analysis will provide insights into the mechanisms that establish the Indonesian–Australian monsoon. Future analysis would also benefit from improvements in the bioclimatic proxy evidence. A

greater number of sites particularly in the arid interior would help validate the model results, where presently the model simulations are the only data points that can provide a representation of the MH palaeoclimate.



**Author Contribution**

AL designed the simulations and performed the analysis. AL prepared the manuscript in consultation with and contributions from HM.

**Competing Interests**

The authors declare that they have no conflict of interest.

**Acknowledgements**

This work was funded by Rock Art Australia and Dunkeld Pastoral Co. Pty Ltd as part of Australian Research Council Linkage Project LP170100242. The simulations were performed using The University of Queensland's Research Computing Centre high performance computing facilities. The bioclimatic proxy dataset was provided by Annika Herbert for which we are grateful. We are grateful for the advice and restart files for the CESM simulations, provided by Jiang Zhu at NCAR.

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
