# Peer review of "Insights into the Australian mid-Holocene climate using downscaled climate models"

_EGUsphere, 2024_

## Author Comment (AC1)

Response to reviewer – RC1

We repeat the reviewer's comments here in black and our response is in blue.

The authors carried out downscaling simulation of the mid-Holocene Australian climate based on the WRF and compared it with the CESM simulation of the mid-Holocene Climate and obtained an added value. The study is certainly very interesting. Therefore, I suggest a minor revision.

1. I see that the authors have also given almost identical distribution of vegetation in the WRF for both the Holocene and control experiments. This is similar to the fact that most PMIP experiments still use the vegetation distribution of present day. Actually, the reconstruction results suggest that the vegetation distribution in the mid-Holocene may have been quite different from the present day and may have impacted on the regional climate (Thompson et al., 2022; Sun et al., 2023). For this reason, I suggest that authors might try to drive vegetation models (Such as Biome et al.) with CESM outputs and WRF outputs in order to examine the regional vegetation response to mid-Holocene climate change.

Thompson, A.J., Zhu, J., Poulsen, C.J., Tierney, J.E., & Skinner, C.B. (2022). Northern Hemisphere vegetation change drives a Holocene thermal maximum. Science Advances, 8.

Sun, Y., Wu, H., Ramstein, G. et al. Revisiting the physical mechanisms of East Asian summer monsoon precipitation changes during the mid-Holocene: a data–model comparison. Clim Dyn 60, 1009–1022 (2023). https://doi.org/10.1007/s00382-022-06359-1

We thank the review for this comment and bringing to our attention these two papers. We agree that vegetation changes between the mid-Holocene and pre-industrial should play a role in analysis of the regional climate. To account for this, the approach in the WRF simulations was to modify the land-use classification from the results of Allen et al. (2020). It transpired, as noted by the reviewer, that Allen et al. (2020)'s results presented a very similar vegetation distribution between the two periods as shown in Fig. 3. This is further corroborated, for example, by vegetation evidence from Rowe et al. (2019) that there has been little change in vegetation type over northern Australia between the mid-Holocene and present.

It could indeed be informative to use the results from our paper to drive an offline vegetation model, but we feel that would be outside the scope of the current manuscript.

2. Besides, in the introduction somewhere, previous progress in downscaling studies on other regions of the mid-Holocene should be reviewed.

Huo, Y., Peltier, W. R., and Chandan, D.: Mid-Holocene monsoons in South and Southeast Asia: dynamically downscaled simulations and the influence of the Green Sahara, Clim. Past, 17, 1645–1664, https://doi.org/10.5194/cp-17-1645-2021, 2021.

We appreciate the link to this article and will include this in the final manuscript.

References

Allen, J. R. M., Forrest, M., Hickler, T., Singarayer, J. S., Valdes, P. J., and Huntley, B.: Global vegetation patterns of the past 140,000 years, J. Biogeogr., 47, 2073–2090, https://doi.org/10.1111/jbi.13930, 2020.

Rowe, C., Brand, M., Hutley, L.B., Wurster, C., Zwart, C., Levchenko, V., Bird, M., 2019. Holocene savanna dynamics in the seasonal tropics of northern Australia. Rev. Palaeobot. Palynol. 267, 17–31. https://doi.org/10.1016/j.revpalbo.2019.05.004

---

## Author Comment (AC2)

Response to reviewer – RC2

We repeat the reviewer's comments here in black and our response is in blue.

Summary

This manuscript assessed the ability of the CESM and WRF to simulate the mid-Holocene climate of Australia and the equatorial tropics of the Indonesia-Australian monsoon regions with respect to bioclimatic modelled proxy data, in terms of temperature, precipitation and plant available water index. This study provides the first downscaled paleoclimate analysis of the mid-Holocene in Australia, and is well written. The followings need to be commented or addressed before it is publishable.

**General comments**

For Introduction

1. Add the description about the need for RCM in paleoclimate modeling and the progress in downscaling for the paleoclimate simulations.

   We thank the reviewer for this comment. The last paragraph of the introduction (lines 54-67) discusses the previous work using RCMs in palaeoclimate modelling and notes the need for this approach due to "better depiction of feedback and physical processes at the regional scale". We feel, considering the reviewer's comment, that this could be strengthened by elaborating on the progress that has been made over the ca. 20 years since the first paper in this field from 2003. We propose to amend lines 60-63 as follows:

   The early work using downscaled climate models employed modifications to the insolation and greenhouse gas concentrations of the mid-Holocene climate, and a resolution of ca. 50 km. As the field has evolved research has begun to interrogate the influence of vegetation as a boundary condition (e.g. Ludwig et al., 2017; Strandberg et al., 2014, 2022), and use available bioclimatic proxy evidence to validate the modelling results (Ludwig et al., 2017; Russo et al., 2024; Strandberg et al., 2022). There is evidence that some improvement in simulated climate can be achieved by using finer resolution models (Armstrong et al., 2019; Ludwig et al., 2017), particularly hydrological processes (Ludwig et al., 2019), and that vegetation plays an important role in the simulation of the palaeoclimate (Strandberg et al., 2022). These improvements stem from the better depiction of feedback and physical processes at the regional scale (Armstrong et al., 2019; Ludwig et al., 2017).

For the Results

1. It is inappropriate to use only one metric, the mean absolute error, to assess the model simulations. As is shown in Fig. 5-6, the differences between MH and piControl over the northern Australian in WRF and CESM show a warm and dry conditions, which is totally disagree with the pollen proxy datasets, and those differences are even more pronounced in WRF. It seems that the downscaling results do not improve the simulations over the Australian.

   We thank the reviewer for this comment. Taking the list of mid-Holocene papers from the introduction of our paper and the list of papers that have used WRF in downscaled palaeoclimate modelling from the methodology (18 + 13, although Yu et al. 2014 occurs in both lists), there are only two papers (Armstrong et al., 2019; Ludwig et al., 2017) that

provide statistics. Armstrong et al. (2019) plot a linear regression to the model-proxy data but do not report any statistics, and furthermore restrict the linear fit to only statistically significant proxy data points. Ludwig et al. (2017) provides the "mean deviation", which is analogous to mean absolute error. There are three other papers (Russo et al., 2024; Velasquez et al., 2021; Paeth et al., 2019) that plot the proxy data overlaying the model results. Ludwig et al. (2021) provides a table of the few proxy data sites and the corresponding values from the model results. There are many still that use the available proxy datasets for their region, but only report domain, or sub-domain temperature differences from the proxy datasets and compare those to the same regions model results (Ludwig et al., 2018; Strandberg et al., 2014, 2022) or provide differences for specific sites (Ludwig and Hochmann 2022).

Furthermore, as many studies have noted there can be considerable spread between the proxy data results, which is the case over Australia as well. Therefore, any statistic that implies the proxy data is the truth to which the model should fit, would not be an accurate assumption.

Consequently, we believe that reporting of a single statistic is informative and exceeds the detail provided in many commensurate papers. We propose to add additional tables to the supplementary material similar to Ludwig et al., (2021) that will increase the evidence base for our results.

Regarding the differences between the proxy dataset and the model results from CESM and WRF, we agree that in some variables there is not an improvement from the use of downscaled modelling, but this does not necessarily detract from the other benefits from downscaled modelling as outlined in the paper. We note that there is large uncertainty in the pollen proxy dataset, which we will include in the proposed supplementary tables, and there are issues with the pollen proxy dataset which we discussed in lines 339-346 in the manuscript.

2. Does the warm-dry biases over the northern Australian in WRF pre-industrial simulation have an impact on the simulation results for MH?

   We thank the reviewer for this comment. There is limited connectivity between the two simulations restricted to factors that transmit from CESM to WRF that are the same in both the CESM pre-industrial and mid-Holocene simulations. These would at most have a miniscule impact on the mid-Holocene WRF simulation.

3. Change the 850 hPa divergence difference in Fig.9 to the Whole-layer water vapor flux divergence difference.

   We thank the reviewer for this suggestion and will amend Fig. 9 as recommended including commentary and reference to this figure within a revised manuscript.

4. Give the possible reason for the differences in annual plant available water index over the Indonesia between WRF and CESM in Fig.10.

   We thank the reviewer for this comment. The differences between WRF and CESM plant available water index over the tropics are primarily related to decreases in potential evapotranspiration during the mid-Holocene in the CESM simulation. We note also that WRF simulated much larger precipitation values compared to CESM, which we noted were more realistic when the WRF and CESM pre-industrial values were compared to

the Tropical Rainfall Monitoring Mission (TRMM) dataset (lines 160-170 in the manuscript). The WRF simulation showed larger positive precipitation anomalies in the mid-Holocene, compared to the CESM simulation, and these are co-located with slightly reduced evapotranspiration.

We will amend the manuscript with to include this detail.

5. Add some discussion about the added value of WRF compared to the driving forces.

We thank the reviewer for this comment and will elaborate on this in the introduction, see for example the above response to Introduction comment #1.

**Specific comments**

What does the dots mean in the Figure S5?

We thank the reviewer for this comment. We will amend the figure caption to read: "Seasonal 10 m wind barbs for …"

References

Armstrong, E., Hopcroft, P. O., and Valdes, P. J.: Reassessing the Value of Regional Climate Modeling Using Paleoclimate Simulations, Geophys. Res. Lett., 46, 12464–12475, https://doi.org/10.1029/2019GL085127, 2019.

Ludwig, P., Pinto, J. G., Raible, C. C., and Shao, Y.: Impacts of surface boundary conditions on regional climate model simulations of European climate during the Last Glacial Maximum, Geophys. Res. Lett., 44, 5086–5095, https://doi.org/10.1002/2017GL073622, 2017.

Ludwig, P., Shao, Y., Kehl, M., and Weniger, G. C.: The Last Glacial Maximum and Heinrich event I on the Iberian Peninsula: A regional climate modelling study for understanding human settlement patterns, Glob. Planet. Change, 170, 34–47, https://doi.org/10.1016/j.gloplacha.2018.08.006, 2018.

Ludwig, P., Gavrilov, M. B., Markovic, S. B., Ujvari, G., and Lehmkuhl, F.: Simulated regional dust cycle in the Carpathian Basin and the Adriatic Sea region during the Last Glacial Maximum, Quat. Int., 581–582, 114–127, https://doi.org/10.1016/j.quaint.2020.09.048, 2021.

Ludwig, P. and Hochman, A.: Last glacial maximum hydro-climate and cyclone characteristics in the Levant: A regional modelling perspective, Environ. Res. Lett., 17, https://doi.org/10.1088/1748-9326/ac46ea, 2022.

Paeth, H., Steger, C., Li, J., Pollinger, F., Mutz, S. G., and Ehlers, T. A.: Comparison of climate change from Cenozoic surface uplift and glacial-interglacial episodes in the Himalaya-Tibet region: Insights from a regional climate model and proxy data, Glob. Planet. Change, 177, 10–26, https://doi.org/10.1016/j.gloplacha.2019.03.005, 2019.

Russo, E., Fallah, B., Ludwig, P., Karremann, M., and Raible, C. C.: The long-standing dilemma of European summer temperatures at the mid-Holocene and other considerations on learning from the past for the future using a regional climate model, Clim. Past, 18, 895–909, https://doi.org/10.5194/cp-18-895-2022, 2022.

Strandberg, G., Kjellström, E., Poska, A., Wagner, S., Gaillard, M. J., Trondman, A. K., Mauri, A.,

Davis, B. A. S., Kaplan, J. O., Birks, H. J. B., Bjune, A. E., Fyfe, R., Giesecke, T., Kalnina, L., Kangur, M., Van Der Knaap, W. O., Kokfelt, U., Kuneš, P., Lataowa, M., Marquer, L., Mazier, F., Nielsen, A. B., Smith, B., Seppä, H., and Sugita, S.: Regional climate model simulations for Europe at 6 and 0.2 k BP: Sensitivity to changes in anthropogenic deforestation, Clim. Past, 10, 661–680, https://doi.org/10.5194/cp-10-661-2014, 2014.

Strandberg, G., Lindström, J., Poska, A., Zhang, Q., Fyfe, R., Githumbi, E., Kjellström, E., Mazier, F., Nielsen, A. B., Sugita, S., Trondman, A. K., Woodbridge, J., and Gaillard, M. J.: Mid-Holocene European climate revisited: New high-resolution regional climate model simulations using pollen-based land-cover, Quat. Sci. Rev., 281, https://doi.org/10.1016/j.quascirev.2022.107431, 2022.

Velasquez, P., Kaplan, J. O., Messmer, M., Ludwig, P., and Raible, C. C.: The role of land cover in the climate of glacial Europe, Clim. Past, 17, 1161–1180, https://doi.org/10.5194/cp-17-1161-2021, 2021.